# SCALABLE LLM MATH REASONING ACCELERATION WITH LOW-RANK DISTILLATION

## ABSTRACT

Due to long generations, large language model (LLM) math reasoning demands significant computational resources and time. While many existing efficient inference methods have been developed with excellent performance preservation on language tasks, they often severely degrade math performance. In this paper, we propose `Caprese`, a resource-efficient distillation method to recover lost capabilities from deploying efficient inference methods, focused primarily in feedforward blocks. With original weights unperturbed, roughly 1% of additional parameters, and only 20K synthetic training samples, we are able to recover much if not all of the reasoning capabilities lost from efficient inference for thinking LLMs and without harm to language tasks for instruct LLMs. Moreover, `Caprese` slashes the number of active parameters (∼2B cut for Gemma 2 9B and Llama 3.1 8B) and integrates cleanly into existing model layers to reduce latency (>16% time-to-next-token reduction) while encouraging response brevity (up to 8.5% fewer tokens).

## 1 INTRODUCTION

With the increasing capabilities of large language models (LLMs) (Vaswani et al., 2017), evaluations are also becoming increasingly sophisticated, typically involving multi-step reasoning such as in math problem solving. These tasks tend to demand long generation which drives up latency, making efficiency a dire issue. Fortunately, many sparsity-based efficient LLM inference algorithms have shown great promise, slashing expensive computational bottlenecks with little damage to the original performance on a variety of language-based tasks like reading comprehension and summarization. *However, for math reasoning tasks, many of these algorithms begin to break down, decimating performance, despite their robustness in language settings.* Thus, there is a need to design efficient inference algorithms that simultaneously maintain language and math capabilities.

One of the main differences between math reasoning and many language tasks is the generation length. Reasoning typically involves long generations from chain-of-thoughts (CoTs) (Wei et al., 2022), which often far surpass the length of the input query. However, CoT performance falls apart with efficient algorithms that introduce approximation errors which build up over time. For instance, deploying CATS (Lee et al., 2024), a sparse thresholding method, on Gemma 2 2B (Team et al., 2024) has little impact on language generation performance, but knocks GSM8K (Cobbe et al., 2021) accuracy *from 51.02% to 34.42%* (Table 1). Similarly, for thinking models: applying GRIFFIN (Dong et al., 2024a), an adaptive structured pruning method, on the first half of DeepSeek-R1-Distill-Qwen 1.5B (Guo et al., 2025) drops MATH-500 (Lightman et al., 2023) accuracy *from 79.40% to 42.00%* (Table 2). Moreover, since generation is much more computationally demanding than prefill, there is a dire need to make long reasoning CoTs efficient, which is made more apparent with the rise of test-time scaling. Thus, this poses two key inference challenges with math reasoning. First, even single token mistakes can sometimes drive the generation trajectory off course, leading to an incorrect response (Zhou et al., 2024). Second, the already computationally expensive LLM autoregressive decoding process is accentuated with generating CoTs. This problem is further exaggerated as CoTs scale in length in thinking models (Guo et al., 2025) and as the number of repeated generations scale to elicit higher quality answers (Brown et al., 2024; Wu et al., 2024; Snell et al., 2024). *An ideal method should be efficient, performance preserving, and easy to integrate.*

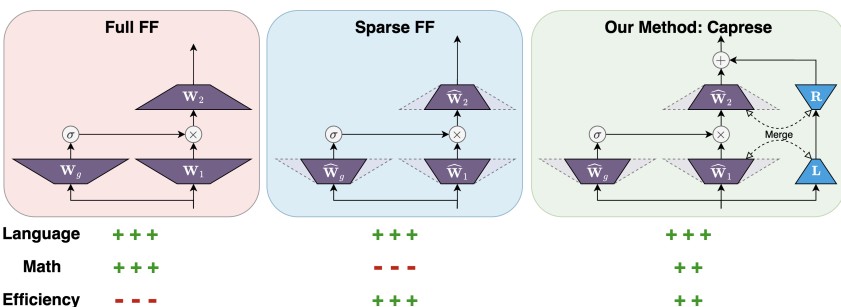

Figure 1: A full FF block maximizes accuracy without any benefit to efficiency. Sparse FF algorithms can be very efficient by using subsets of the FF block but harm math performance. Our method, `Caprese`, uses a sparse FF algorithm and a small distilled low-rank linear layer, which can be merged with existing FF weights, for performative inference in language and math settings while being efficient. Layers are drawn as trapezoids to highlight the relative size of inputs and outputs.

Thankfully, low-rank structure in the feedforward (FF) output features can help make this possible. (We choose to focus on FF blocks since they contribute around 2/3 of an LLM's parameters and about 50% of the generation latency.) Because of the success of works that exploit contextual sparsity in FF blocks on language tasks for efficiency, like CATS and GRIFFIN, we peek into the residuals of FF sparsity-based efficient methods. Using an oracle top-$k$ filter on FF nonlinearity output magnitudes, we observe huge reductions in error with a low-rank approximation to the FF output residuals (Figure 2). Since FF intermediate feature sizes can be on the order of $10^5$, adding 256 for a low-rank approximation is comparatively tiny. *This observation motivates us to estimate the residual from sparse FF methods with low-rank layers.*

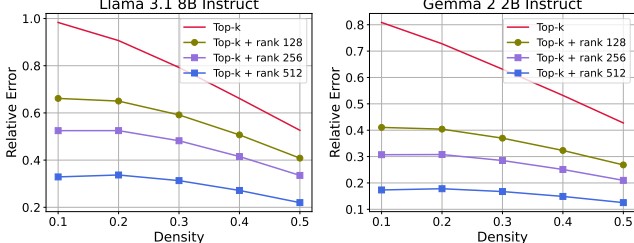

Figure 2: Average relative FF output error of generated tokens with varying top-$k$ densities and low-rank approximations. The density is the fraction of non-zero intermediate FF features maintained by top-$k$. Samples consist of 16 random MATH generations by the original model. *A relatively small low-rank approximation to the top-$k$ residual reduces error more effectively than increasing density.*

We introduce **`Caprese`** (**CAP**ability **RE**covery with **S**calable **E**fficiency) to learn FF residuals from *any* sparse FF algorithm using small low-rank linear layers for improved performance (Figure 1). While `Caprese` works for any task, we focus on math in this paper, as current efficient sparse methods can obliterate an LLM's math capabilities. Through distillation of math knowledge into these low-rank layers, `Caprese` is able to overcome the aforementioned challenges of reasoning and makes significant progress towards an ideal efficient method:

1. **Performance Enhancement:** Demonstrated across a variety of instruct and thinking models, `Caprese` *recovers much if not all of the math performance lost from deploying a sparse FF algorithm without harming language tasks*. Also, `Caprese` maintains its performance benefit when applying different techniques of test-time scaling.

2. **Efficiency:** Due to its parallelizability with existing FF layers, `Caprese` adds negligible overhead, preserving latency reductions of the underlying method.

3. **Low Budget:** The distillation process of `Caprese` is cheap since we are able to see significant gains in math performance by training on only 20K synthetic math samples. Moreover, with a low-rank layer size of only 256, this equates to adding roughly $0.8\%$ additional parameters into Llama 3.1 8B and Gemma 2 9B, dwarfed by savings in active parameters ($\sim$2B cut for both models with `Caprese` (CATS)).

Our extensive experiments on `Caprese` indicate strong performance on various tasks and models. For example, DeepSeek-R1-Distill-Qwen 7B with a sparse method drops AMC 2023 accuracy from 75.00% to 62.50%, but `Caprese` is able to bring it to 78.13%, beyond the original accuracy. Furthermore, scaling generation quantity with `Caprese` on Llama 3.2 3B Instruct improves Pass@100 by 7.0% from the original model while using 15.8% fewer active parameters. We also show `Caprese` reduces latency by 16% using the Qwen 2.5 14B architecture.

**Paper Organization.** Section 2 covers background on related works and describes GRIFFIN and CATS, two efficient sparse FF algorithms that serve as baselines. Section 3 details `Caprese`'s architecture and distillation procedure. Then, we showcase the strong performance of `Caprese` in Section 4: instruct LLM inference (Section 4.1), scaling generation outputs (Section 4.2), and scaling generation length with thinking LLMs (Section 4.3). Finally, we quantify efficiency improvements in Section 4.4.

## 2 BACKGROUND

In this section, we provide an overview of the efficient LLM inference literature that `Caprese` builds upon with in-depth descriptions of GRIFFIN and CATS which we use as baselines and some recent developments for test-time scaling.

### 2.1 RELATED WORKS

**Efficient Inference for FF Blocks.** To improve the efficiency of FF blocks in LLMs, various methods leverage existing sparse structures in FF features (Geva et al., 2020; Dettmers et al., 2022; Li et al., 2022; Dong et al., 2023; Liu et al., 2023). Pruning sets a parameter subset to zero to enforce static sparsity (LeCun et al., 1989; Sun et al., 2023; Frantar & Alistarh, 2023; Ma et al., 2023). Mixtures of experts (MoEs) adaptively select predefined parameter subsets in FF blocks, though they tend to require significant fine-tuning or training from scratch (Jacobs et al., 1991; Zhang et al., 2021; Dai et al., 2024; Liu et al., 2023). Similar to MoEs, another line of work aims at exploiting contextual sparsity to dynamically select FF neurons for each input without training. GRIFFIN (Dong et al., 2024a) is a calibration-free method that uses FF activations from the prefill phase to adaptively prune FF neurons for generation. CATS (Lee et al., 2024) uses hard thresholding to skip computation in parts of the FF block with a custom kernel.

**Shorter CoTs.** Several works aim to improve reasoning efficiency by directly reducing the number of generated tokens (Renze & Guven, 2024; Nayab et al., 2024; Arora & Zanette, 2025; Aytes et al., 2025; Xu et al., 2025; Qu et al., 2025). Since we aim to reduce the per-token latency, this line of research to encourage brevity is orthogonal to ours and could be used alongside our method.

**Test-time Scaling.** To enhance LLM performance, the priority has historically been to scale training with more data and bigger models (Kaplan et al., 2020; Hoffmann et al., 2022). More recently, there is an increasing effort towards test-time scaling to boost performance on difficult tasks like math. Two ways of test-time scaling that have seen great success. First, for a single prompt, multiple responses can be sampled from the LLM (Brown et al., 2024; Snell et al., 2024; Wu et al., 2024; Manvi et al., 2024; Sun et al., 2024). This increases the probability that a desired response lies in the pool of responses. While this method of scaling is highly parallelizable, it also relies on a verifier model to select the best one. Second, CoTs can be lengthened for each response (Guo et al., 2025; Muennighoff et al., 2025; Yang et al., 2025). By extrapolating the success of CoTs to extreme lengths (e.g., DeepSeek-R1 generates up to 32K tokens per response), the accuracy of the final answer improves significantly, but this method has low parallelizability due to the autoregressive nature of generation.

### 2.2 TRAINING-FREE & ADAPTIVELY SPARSE FF METHODS

We now briefly describe the inner workings of GRIFFIN and CATS. Let $\boldsymbol{X} \in \mathbb{R}^{S \times D}$ be the input into the FF block during the prefill phase with sequence length $S$ and feature size $D$. Define the FF block as $\mathrm{FF}(\boldsymbol{X}) = \mathrm{FF}_2(\mathrm{FF}_1(\boldsymbol{X}))$ such that

$$\boldsymbol{Z} = \mathrm{FF}_1(\boldsymbol{X}) = \sigma(\boldsymbol{X}\boldsymbol{W}_g) \odot \boldsymbol{X}\boldsymbol{W}_1, \tag{1}$$

$$\text{FF}_2(\boldsymbol{Z}) = \boldsymbol{Z}\boldsymbol{W}_2, \tag{2}$$

where $\boldsymbol{W}_g, \boldsymbol{W}_1 \in \mathbb{R}^{D \times D_{\text{FF}}}$, $\boldsymbol{W}_2 \in \mathbb{R}^{D_{\text{FF}} \times D}$, $\sigma$ is a nonlinear function, $\odot$ is an element-wise multiplication operator, and $D_{\text{FF}}$ is the FF intermediate feature size. Typically, $D_{\text{FF}} \gg D$. Although recent LLMs use this architecture, for LLMs without gated functions (e.g., OPT (Zhang et al., 2022)), $\boldsymbol{X}\boldsymbol{W}_1$ can be removed. Bias terms are omitted for brevity.

**GRIFFIN.** GRIFFIN (Dong et al., 2024a) adaptively prunes columns and rows in the FF block weights, using FF activation statistics from the prefill phase, namely the flocking patterns. GRIFFIN calculates $[\overline{\boldsymbol{Z}}]_i = [\boldsymbol{Z}]_i / \|[\boldsymbol{Z}]_i\|_2$ for each token index $i$, followed by an aggregation across the FF feature axis: $[\boldsymbol{s}]_j = \|[\overline{\boldsymbol{Z}}]_{\cdot,j}\|_2$. The result $\boldsymbol{s} \in \mathbb{R}^{D_{\text{FF}}}$ gives a metric to perform top-$k$ selection across corresponding columns of $\boldsymbol{W}_g$ and $\boldsymbol{W}_1$, and rows of $\boldsymbol{W}_2$ to produce $\widehat{\boldsymbol{W}}_g, \widehat{\boldsymbol{W}}_1 \in \mathbb{R}^{D \times k}$, and $\widehat{\boldsymbol{W}}_2 \in \mathbb{R}^{k \times D}$. Then, for the generation phase, the following FF block is used for input $\boldsymbol{x} \in \mathbb{R}^D$:

$$\widehat{\boldsymbol{z}} = \widehat{\text{FF}}_1(\boldsymbol{x}) = \sigma(\boldsymbol{x}\widehat{\boldsymbol{W}}_g) \odot \boldsymbol{x}\widehat{\boldsymbol{W}}_1, \tag{3}$$

$$\widehat{\text{FF}}_2(\widehat{\boldsymbol{z}}) = \widehat{\boldsymbol{z}}\widehat{\boldsymbol{W}}_2. \tag{4}$$

The compressed FF blocks are fixed throughout generation but are dynamic across prompts.

**CATS.** CATS (Lee et al., 2024) uses hard thresholding to skip computation in part of the FF block. Letting $T_\tau$ be the hard thresholding function with threshold $\tau$, CATS computes

$$\widehat{\boldsymbol{z}} = \widehat{\text{FF}}_1(\boldsymbol{x}) = T_\tau(\sigma(\boldsymbol{x}\boldsymbol{W}_g)) \odot \boldsymbol{x}\boldsymbol{W}_1, \tag{5}$$

$$\widehat{\text{FF}}_2(\widehat{\boldsymbol{z}}) = \widehat{\boldsymbol{z}}\boldsymbol{W}_2. \tag{6}$$

The weights $\boldsymbol{W}_1$ and $\boldsymbol{W}_2$ should be sparsified into $\widehat{\boldsymbol{W}}_1$ and $\widehat{\boldsymbol{W}}_2$, respectively, based on the non-zero entries of $T_\tau(\sigma(\boldsymbol{x}\boldsymbol{W}_g))$ for latency improvement, which can vary from token to token and require a custom kernel for wall clock speed-up. The parameter $\tau$ is calibrated to be a desired percentile on a dataset. In this paper, to avoid calibration, we set $\tau$ based on prefill features and only threshold during the generation phase, analogous to GRIFFIN.

## 3 METHOD: CAPRESE

Motivated by the low-rank structure of residuals in Figure 2, we introduce `Caprese` which distills approximation errors in embeddings into low-rank linear layers in FF blocks. See Figure 1 for an illustration of our method.

### 3.1 LAYER-WISE DISTILLATION

Inference efficiency algorithms often introduce feature approximation errors in favor of faster generation, which we mitigate with distillation. Let the efficient and approximate FF block be $\widehat{\text{FF}}(\boldsymbol{x}) = \widehat{\text{FF}}_2(\widehat{\text{FF}}_1(\boldsymbol{x}))$. In our design of `Caprese`, we do not constrain $\widehat{\text{FF}}$ to be any specific method or architecture. For instance, $\widehat{\text{FF}}$ could be an FF block with GRIFFIN (Dong et al., 2024a) or CATS (Lee et al., 2024). Then, the error is

$$\|\text{FF}(\boldsymbol{x}) - \widehat{\text{FF}}(\boldsymbol{x})\|_2^2.$$

We choose to reduce this residual with a low-rank linear layer, meaning we want to solve

$$\min_{\boldsymbol{L}, \boldsymbol{R}} \frac{1}{|\mathcal{X}|} \sum_{\boldsymbol{x} \in \mathcal{X}} \|\text{FF}(\boldsymbol{x}) - \widehat{\text{FF}}(\boldsymbol{x}) - \boldsymbol{x}\boldsymbol{L}\boldsymbol{R}\|_2^2 \tag{7}$$

for input set $\mathcal{X}$, $\boldsymbol{L} \in \mathbb{R}^{D \times r}$, and $\boldsymbol{R} \in \mathbb{R}^{r \times D}$ where $r \ll D_{\text{FF}}$. The optimal solution can be computed analytically since this is a reduced rank regression problem, but the size of $|\mathcal{X}|$ and $D$ may make it prohibitively expensive. Therefore, we opt to learn $\boldsymbol{L}$ and $\boldsymbol{R}$ independently for every FF block (i.e., previous FF blocks are assumed to be from the original model), allowing for parallel layer-wise training. This takes inspiration from LESS (Dong et al., 2024b) which uses layer-wise training on attention residuals for key-value cache compression. Each $\boldsymbol{R}$ is initialized as a zero matrix since the efficient approximation is assumed to be of good quality, and original model weights are frozen. We also distill end-to-end (E2E) to further improve the performance.

## 3.2 END-TO-END DISTILLATION

Using the learned low-rank layers as an initialization, we put them all together to distill the final model embedding before the linear head into the efficient model, again using MSE:

$$\min_{(\boldsymbol{L}_i, \boldsymbol{R}_i)_{i=1,\dots,L}} \frac{1}{|\mathcal{X}|} \sum_{\boldsymbol{x} \in \mathcal{X}} \|\mathrm{M}(\boldsymbol{x}) - \mathrm{M}_{\text{student}}(\boldsymbol{x})\|_2^2 \tag{8}$$

where M and $\mathrm{M}_{\text{student}}$ are the original LLM and efficient LLM with distillation layers, respectively, excluding the final linear head. $L$ is the number of layers in the model. All original weights are frozen, so the only tunable parameters are the $\boldsymbol{L}$ and $\boldsymbol{R}$ of each FF block.

## 3.3 PARALLEL INFERENCE COMPUTATION

The computation of $\boldsymbol{x}\boldsymbol{L}\boldsymbol{R}$ can be done in parallel with the original FF operations. In fact, $\boldsymbol{L}$ and $\boldsymbol{R}$ can be concatenated with the up and down projection matrices, respectively. In other words, the `Caprese` FF block is $\widehat{\mathrm{FF}}^+(\boldsymbol{x}) = \widehat{\mathrm{FF}}_2^+(\widehat{\mathrm{FF}}_1^+(\boldsymbol{x}))$, such that

$$\widehat{\boldsymbol{z}}^+ = \widehat{\mathrm{FF}}_1^+(\boldsymbol{x}) = \left[ \sigma(\boldsymbol{x}\widehat{\boldsymbol{W}}_g) \quad \mathbf{1}_r \right] \odot \boldsymbol{x} \left[ \widehat{\boldsymbol{W}}_1 \quad \boldsymbol{L} \right], \tag{9}$$

$$\widehat{\mathrm{FF}}_2^+(\widehat{\boldsymbol{z}}^+) = \widehat{\boldsymbol{z}}^+ \left[ \widehat{\boldsymbol{W}}_2^\top \quad \boldsymbol{R}^\top \right]^\top, \tag{10}$$

where $\mathbf{1}_r$ is a one-vector with length $r$. In practice, to save memory and time, we do not materialize $\mathbf{1}_r$ but directly assign the product with $\sigma(\boldsymbol{x}\widehat{\boldsymbol{W}}_g)$ to corresponding entries of $\boldsymbol{x}\widehat{\boldsymbol{W}}_1^+$. Recall that the prefill stage still just uses the original model.

## 3.4 TRAINING DETAILS

We set the inner dimension of our low-rank layer to $r = 256$ (to see the effect of different $r$, see Appendix A). In comparison to the large inner dimension of FF layers (e.g., $D_{\text{FF}} = 14336$ for Llama 3.1 8B and Gemma 2 9B), our choice of $r$ is relatively miniscule, adding only $\sim 1\%$ new parameters for all tested models (Appendix B). We use a 20K subset of a common synthetic math training set for training. For a fair comparison, the same subset is used for both layer-wise and E2E distillation for every model. Each training sample is prepended with a CoT instruction: "Please reason step by step." At test time, the actual instructions may be vastly different. Layer-wise and E2E training consists of 20 epochs and 3 epochs, respectively. Training and inference are done in BF16. Hyperparameters are listed in Appendix G.

## 3.5 COMPARISON WITH LORA

Our method derives inspiration from and shares a slight connection with low-rank adaptation (LoRA), a widely used method for efficient fine tuning with the addition of low-rank parameters (Hu et al., 2022), but remains distinct since they target different inefficiencies. To illustrate, recall a sparse algorithm constructs $\widehat{\boldsymbol{W}}_1 \in \mathbb{R}^{D \times k}$ by selecting columns from $\boldsymbol{W}_1$. Alternatively, construct $\widetilde{\boldsymbol{W}}_1 \in \mathbb{R}^{D \times D_{\text{FF}}}$ by setting unwanted columns to zero, so nonzero columns in $\widetilde{\boldsymbol{W}}_1$ match columns in $\widehat{\boldsymbol{W}}_1$ and vice versa. Doing the same to columns of $\boldsymbol{W}_g$ and rows of $\boldsymbol{W}_2$, LoRA learns $\boldsymbol{A}_1, \boldsymbol{A}_g, \boldsymbol{B}_2^\top \in \mathbb{R}^{D \times r}$ and $\boldsymbol{B}_1, \boldsymbol{B}_g, \boldsymbol{A}_2^\top \in \mathbb{R}^{r \times D_{\text{FF}}}$ to form a new FF block with LoRA weights:

$$\widetilde{\mathrm{FF}}(\boldsymbol{x}) = \left( \sigma\big(\boldsymbol{x}(\widetilde{\boldsymbol{W}}_g + \boldsymbol{A}_g\boldsymbol{B}_g)\big) \odot \big(\boldsymbol{x}(\widetilde{\boldsymbol{W}}_1 + \boldsymbol{A}_1\boldsymbol{B}_1)\big) \right) (\widetilde{\boldsymbol{W}}_2 + \boldsymbol{A}_2\boldsymbol{B}_2). \tag{11}$$

From this, we see *LoRA is not designed to be applied to sparse efficient inference algorithms*. First, this is because in its compressed form, LoRA requires two sequential operations per linear layer. In contrast, `Caprese` parameters can be appended to existing ones, so additional computation is all parallel to the original operations (Section 3.3). The real latency impact of this difference is quantified later in Table 4. `Caprese`'s architecture also allows efficient training, but that is merely a byproduct of our design, not its purpose. Second, for a method like GRIFFIN which reduces the intermediate FF feature size from $D_{\text{FF}}$ to $k$, adding LoRA will negate this benefit while `Caprese` will preserve the reduced feature size. Due to this difference, we de-prioritize comparisons with LoRA in the main text, but include comparisons with LoRA in Appendix C for interested readers.

Table 1: Instruct models' 0-shot accuracies on mathematical reasoning (GSM8K and MATH) and language generation tasks (CoQA, QASPER, XSum, and CNN/DailyMail). FF sparsity is set to 50%, and $r = 256$.

| Model | GSM8K | MATH | CoQA | QASPER | XSum | CNN/DailyMail |
|---|---|---|---|---|---|---|
| *Llama 3.2 1B Instruct* | 22.44 | 10.66 | 55.43 | 14.43 | 21.65 | 25.60 |
| GRIFFIN | 7.13 | 5.42 | 56.05 | 14.11 | 21.13 | 25.47 |
| Layer-wise `Caprese` | 13.72 | 6.62 | 56.07 | 13.40 | 20.65 | 26.18 |
| E2E `Caprese` | **21.00** | **8.44** | 56.55 | 13.88 | 20.71 | 26.18 |
| CATS | 19.18 | 7.54 | 54.40 | 13.88 | 21.10 | 24.73 |
| Layer-wise `Caprese` | 18.65 | 8.28 | 55.58 | 14.35 | 20.94 | 25.35 |
| E2E `Caprese` | **20.39** | **9.04** | 56.12 | 13.75 | 20.40 | 25.56 |
| *Llama 3.2 3B Instruct* | 51.55 | 14.32 | 63.95 | 12.45 | 23.22 | 26.20 |
| GRIFFIN | 28.96 | 10.98 | 64.52 | 12.52 | 22.09 | 25.49 |
| Layer-wise `Caprese` | 40.18 | 13.70 | 64.33 | 11.60 | 21.56 | 25.90 |
| E2E `Caprese` | **44.66** | **16.96** | 64.83 | 12.35 | 21.26 | 26.04 |
| CATS | 41.24 | 12.04 | 58.87 | 11.36 | 22.23 | 25.40 |
| Layer-wise `Caprese` | 45.49 | 13.62 | 60.53 | 12.26 | 22.50 | 25.90 |
| E2E `Caprese` | **46.85** | **14.40** | 61.72 | 12.36 | 21.58 | 25.40 |
| *Gemma 2 2B Instruct* | 51.02 | 16.06 | 63.77 | 10.96 | 22.17 | 26.01 |
| GRIFFIN | 33.74 | 11.32 | 63.28 | 11.07 | 18.27 | 22.24 |
| Layer-wise `Caprese` | 42.53 | 12.32 | 63.77 | 10.75 | 21.63 | 26.37 |
| E2E `Caprese` | **48.14** | **13.70** | 63.37 | 11.05 | 22.48 | 27.16 |
| CATS | 34.42 | 10.56 | 61.53 | 10.11 | 21.97 | 26.46 |
| Layer-wise `Caprese` | **46.32** | 13.90 | 61.92 | 10.82 | 22.10 | 26.23 |
| E2E `Caprese` | 46.17 | **14.16** | 63.92 | 11.03 | 22.15 | 26.52 |

## 4 EXPERIMENTS

We showcase the effectiveness of `Caprese` at recovering much, if not all, of the reasoning performance lost from efficient inference algorithms without sacrificing efficiency or performance on language tasks. In the following section (Section 4.1), we observe better math performance for instruct LLMs without losing their generalizability. Then, we study `Caprese`'s benefit on two axes of test-time scaling: number of generations (Section 4.2) and generation length with thinking models (Section 4.3). Finally, we highlight `Caprese`'s latency and length improvements in Section 4.4. When ambiguous, we denote `Caprese` (CATS) to be our method with CATS as the underlying sparse method and similarly for GRIFFIN. Otherwise, we use "`Caprese`" for brevity. Unless specified, the FF intermediate feature sparsity is set at 50%. *NOTE: For a more meaningful baseline, we only apply GRIFFIN to the first half of the model for all experiments since we observed much steeper drops in math performance if used for all layers. CATS is applied to all layers.*

### 4.1 INSTRUCT MODELS

We first look into different instruct LLMs which are performative on non-reasoning tasks and have decent reasoning performance. We show `Caprese` is able to preserve math performance without sacrificing quality on pure language tasks like question answering. We test on the Llama 3 family (Dubey et al., 2024) and Gemma 2 family (Team et al., 2024) of models on 0-shot GSM8K (Cobbe et al., 2021), MATH (Hendrycks et al., 2021), CoQA (Reddy et al., 2019), QASPER (Dasigi et al., 2021), XSum (Narayan et al., 2018), and CNN/DailyMail (Hermann et al., 2015; See et al., 2017). We use CoT prompts for math tasks.

Table 1 shows that *`Caprese` is able to preserve most if not all of the math capabilities in the original models without damaging performance on the language tasks, despite the distillation dataset being all math.* In most cases, CATS and GRIFFIN severely harm GSM8K and MATH accuracy, but `Caprese` is able to effectively recover the lost performance. E2E `Caprese` is the most performative in the majority of math scenarios, suggesting a benefit beyond minimizing local error. `Caprese`'s performance is consistent at different sparsity levels and $r$ (Appendix A). The best performers for CoQA and QASPER are a toss-up, but all methods have little impact on the accuracy

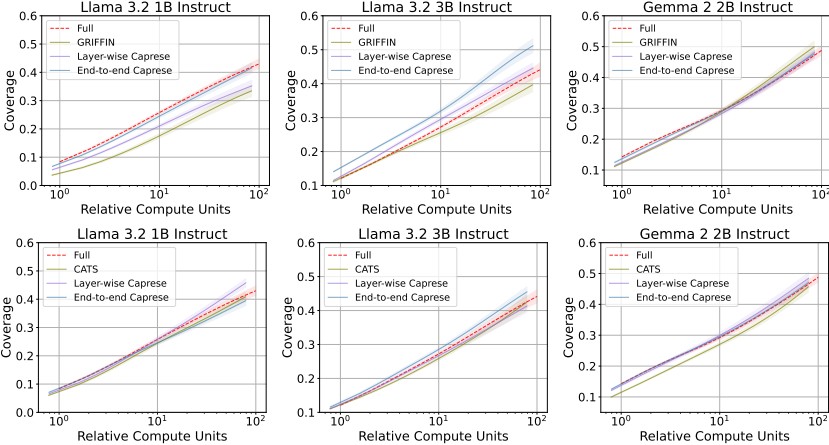

Figure 3: Coverage and standard deviation of 140 samples from MATH as the number of generation attempts, $N$, scales. We define Relative Compute Units $= N \times A$ where $A$ is the fraction of total parameters activated per input. FF sparsity set to 50%. Best viewed zoomed.

of these tasks (the main purpose of these tasks is to show no degradation in language-related cases). Similar benefits with larger instruct models are shown in Table 6, Appendix D.

## 4.2    SCALING BEST-OF-$N$: COVERAGE

Now, we see the generalizability of `Caprese` on the first axis of test-time scaling: sampling multiple responses and selecting the best one, known as Best-of-$N$. Increasing the number of generations per prompt, $N$, is one way to scale inference compute for improved performance, as the probability of generating a correct answer increases. To evaluate, we find the coverage (Pass@$K$) on 140 samples from MATH. We use an oracle verifier to accurately assess the quality of the pool of generated responses. To combat the high variance when $K$ is close to $N$, we calculate the average coverage for $K = 1, \ldots, 100$ across 10 independent pools of 100 generations for each sample. *Factoring in the saved compute, E2E `Caprese` is able to have similar or better coverage scaling compared to the original model*, as shown in Figure 3. Notably, E2E `Caprese` (GRIFFIN) on Llama 3.2 3B Instruct improves Pass@100 by 7.0% from the original model while using 15.8% less compute.

## 4.3    SCALING LENGTH: THINKING MODELS

Thinking models provide another axis of test-time scaling by augmenting the CoT length before giving a final answer. Because this entails very long generation lengths, it is critical that error accumulation across tokens be minimized. We test on DeepSeek-R1-Distill-Qwen (which we abbreviate to DS-Qwen) models (Yang et al., 2024; Guo et al., 2025) using the prompt templates and configurations from Open R1 (Face, 2025) (temperature and top-$p$ set to 0.6 and 0.95, respectively). For these models, we test 0-shot performance on MATH-500 (Hendrycks et al., 2021; Lightman et al., 2023), AIME 2024 (AIME, 2025), AMC 2023 (AMC, 2023), BRUMO 2025 (BRUMO, 2025), and GPQA (Rein et al., 2024), a science benchmark. for a max generation length of 32768. Due to the small size of AIME 2024, AMC 2023, and BRUMO 2025, we evaluate them 4 times, reporting the average accuracy and sample standard deviation.

From Table 2, `Caprese` is the most performative in most cases. DS-Qwen 1.5B and 7B see the greatest benefit with E2E `Caprese` usually achieving the highest accuracy, sometimes even exceeding the full model's performance. DS-Qwen 7B with CATS brings AMC 2023 accuracy down to 62.50% from 75.00%, but `Caprese` lifts performance above the original model's to 78.13%, and similarly with DS-Qwen 7B with CATS and BRUMO 2025. All methods are more robust as the model size increases. For instance, CATS and GRIFFIN had very little impact on DS-Qwen 14B's accuracy on MATH-500, with `Caprese` performing similarly. AIME 2024 and BRUMO 2025 challenge CATS and GRIFFIN with several degradation and partial recovery using our method. *In the next section, we show a simple way to push this recovery even further with reselection.*

Table 2: Thinking models' 0-shot accuracies on reasoning tasks. Sparsity is set at 50%, and $r = 256$. AIME 2024, AMC 2023, and BRUMO 2025 columns also include the sample standard deviation across 4 runs. Further improvements with reselection are shown in Table 3.

| Model | MATH-500 | AIME 2024 | AMC 2023 | BRUMO 2025 | GPQA |
|---|---|---|---|---|---|
| *DS-Qwen 1.5B* | 79.40 | 30.00 $_{\pm 2.72}$ | 60.83 $_{\pm 7.47}$ | 25.83 $_{\pm 3.19}$ | 18.69 |
| GRIFFIN | 42.00 | 2.50 $_{\pm 1.67}$ | 16.88 $_{\pm 1.25}$ | 1.67 $_{\pm 1.92}$ | 11.62 |
| Layer-wise `Caprese` | 47.20 | 2.50 $_{\pm 1.67}$ | 28.13 $_{\pm 2.39}$ | 0.00 $_{\pm 0.00}$ | 13.13 |
| E2E `Caprese` | **60.40** | **6.67** $_{\pm 2.72}$ | **34.38** $_{\pm 6.57}$ | **4.17** $_{\pm 1.67}$ | **16.67** |
| CATS | 72.00 | 11.67 $_{\pm 6.94}$ | 37.50 $_{\pm 3.54}$ | 10.83 $_{\pm 4.19}$ | 11.62 |
| Layer-wise `Caprese` | 73.80 | 15.83 $_{\pm 3.19}$ | **48.13** $_{\pm 3.75}$ | 13.33 $_{\pm 2.72}$ | 16.67 |
| E2E `Caprese` | **74.80** | **20.83** $_{\pm 1.67}$ | 47.50 $_{\pm 3.54}$ | **20.00** $_{\pm 2.72}$ | **22.22** |
| *DS-Qwen 7B* | 90.20 | 47.50 $_{\pm 7.87}$ | 75.00 $_{\pm 4.08}$ | 45.83 $_{\pm 4.19}$ | 38.38 |
| GRIFFIN | 80.60 | 21.67 $_{\pm 4.30}$ | 62.50 $_{\pm 8.90}$ | 25.00 $_{\pm 8.39}$ | 21.72 |
| Layer-wise `Caprese` | 84.80 | 29.17 $_{\pm 3.19}$ | 64.38 $_{\pm 7.18}$ | 27.50 $_{\pm 4.19}$ | **27.78** |
| E2E `Caprese` | **85.40** | **30.00** $_{\pm 6.09}$ | **71.88** $_{\pm 5.54}$ | **29.17** $_{\pm 5.69}$ | 23.74 |
| CATS | 89.20 | 34.17 $_{\pm 7.88}$ | 62.50 $_{\pm 2.04}$ | 37.50 $_{\pm 7.39}$ | 32.83 |
| Layer-wise `Caprese` | 87.00 | **35.00** $_{\pm 4.30}$ | 70.00 $_{\pm 10.21}$ | 39.17 $_{\pm 3.19}$ | **38.38** |
| E2E `Caprese` | **90.00** | 33.33 $_{\pm 4.71}$ | **78.13** $_{\pm 5.15}$ | **47.50** $_{\pm 5.00}$ | 35.86 |
| *DS-Qwen 14B* | 92.80 | 63.33 $_{\pm 2.72}$ | 90.63 $_{\pm 2.39}$ | 60.83 $_{\pm 4.19}$ | 52.53 |
| GRIFFIN | 89.80 | 32.50 $_{\pm 7.39}$ | 80.63 $_{\pm 3.75}$ | 33.33 $_{\pm 3.85}$ | 41.92 |
| Layer-wise `Caprese` | **90.80** | **41.67** $_{\pm 5.77}$ | **86.88** $_{\pm 4.27}$ | **43.33** $_{\pm 4.71}$ | **55.05** |
| E2E `Caprese` | 89.20 | 39.17 $_{\pm 9.18}$ | 84.38 $_{\pm 6.57}$ | 40.00 $_{\pm 2.72}$ | 43.43 |
| CATS | **92.80** | 58.34 $_{\pm 1.92}$ | **88.75** $_{\pm 1.44}$ | **55.00** $_{\pm 3.33}$ | 44.95 |
| Layer-wise `Caprese` | 91.00 | **61.67** $_{\pm 5.77}$ | **88.75** $_{\pm 1.44}$ | 53.00 $_{\pm 7.20}$ | 50.51 |
| E2E `Caprese` | 92.00 | 58.34 $_{\pm 9.62}$ | 87.50 $_{\pm 2.04}$ | 49.17 $_{\pm 4.19}$ | **51.01** |

Table 3: E2E `Caprese` AMC 2023 accuracies and standard deviations when recalculating GRIFFIN pruning metrics and reselecting pruned neurons every $\rho$ decode steps. No reselection and the full model are special cases where $\rho = \infty$ and $\rho = 0$, respectively.

| Model | No Reselect | $\rho = 1024$ | $\rho = 256$ | $\rho = 64$ | Full |
|---|---|---|---|---|---|
| DS-Qwen 1.5B | 34.38 | 41.87 | 45.00 | **58.75** | 60.83 |
| DS-Qwen 7B | 71.88 | 69.25 | 73.13 | **73.75** | 75.00 |
| DS-Qwen 14B | 84.38 | 88.13 | 88.75 | **91.88** | 90.63 |

### 4.3.1 ENHANCED PERFORMANCE WITH RESELECTION

We can push the performance of `Caprese` with neuron reselection. For a sample, GRIFFIN and CATS calculate metrics ($s$ and $\tau$) to determine subsets of the FF block to use, but these metrics are fixed during generation. Updating them mid-generation can benefit downstream performance.

For GRIFFIN, updating the metric $s$ entails integrating the FF feature statistics of generated tokens into $s$. While this can be done by re-running prefill on all tokens, there is a more efficient way by passing in the generated tokens following the last reselection through the full model. As these tokens propagate through each layer, we find the selection metric for the generated tokens $s_G$ and update corresponding KV pairs. This is like verification in speculative decoding (Chen et al., 2023; Leviathan et al., 2023). As $s$ and $s_G$ are $\ell_2$-norms along the token axis, we define the updated metric as $\sqrt{(s \odot s) + (s_G \odot s_G)}$ and use that to reselect different subsets of the FF block to use (as described in Section 2). *This updates the pruned layers yet avoids prefill for all tokens.* Table 3 shows a clear benefit of reselection (even if infrequent compared to speculative decoding) in `Caprese` by pushing the performance much closer to the full model's as we decrease the number of decode steps between reselection rounds. Although GRIFFIN is generally more harmful to accuracy than CATS due to its structured pruning, paired with `Caprese` and reselection, `Caprese` (GRIFFIN) can exceed the performance of `Caprese` (CATS) and reach the full model performance.

Table 4: End-to-end generation latency (s) and average time to next token (ms) for GRIFFIN, LoRA, and `Caprese` using Qwen 2.5 14B. For the "Setup" column, $P + G$ indicates input and generation lengths of $P$ and $G$ tokens, respectively. As before, GRIFFIN is applied to the first half of the model, sparsity is 50%, and $r = 256$.

| Setup | End-to-end Latency (s) | | | | Avg. Time to Next Token (ms) | | | |
|---|---|---|---|---|---|---|---|---|
| | Full | GRIFFIN | LoRA | `Caprese` | Full | GRIFFIN | LoRA | `Caprese` |
| 2048+256 | 10.5 | 8.7 | 9.5 | 8.7 | 41 | 34 | 37 | 34 |
| 2048+2048 | 84.4 | 70.3 | 76.5 | 70.4 | 41 | 34 | 37 | 34 |
| 2048+8192 | 344.9 | 287.7 | 312.5 | 288.8 | 42 | 35 | 38 | 35 |

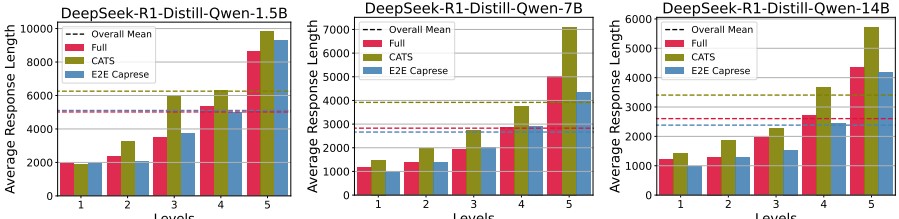

Figure 4: Average number of response tokens for MATH-500 queries with increasing problem difficulty. The global averages are indicated by the dashed lines. Sparsity is set at 50%.

Reselection is also possible with CATS but requires recomputing prefill for all tokens. The parameter to update is the hard thresholding parameter $\tau$, but because this requires finding the desired percentile of intermediate values in the FF block, we would need to access to all values, which likely cannot be fully stored in memory. In turn, prefill will need to be redone anytime we want to update $\tau$. Since $\tau$ represents a cutoff for a desired percentile (e.g., median), it is fairly robust to new observations. Given this and the lack of computational incentive, we focus primarily on reselection for GRIFFIN.

## 4.4 EFFICIENCY

`Caprese` reduces generation latency. Table 4 shows the latencies of different generation settings. `Caprese` cuts total latency and time to next token by >16% for the Qwen 2.5 14B architecture. Moreover, the latency differences between GRIFFIN and `Caprese` are exceptionally small, suggesting that `Caprese` has *minimal overhead*. We also include latencies of GRIFFIN paired with LoRA weights, which although faster than the base model, negates >40% of the time savings that GRIFFIN has provided. This highlights the efficiency suboptimality of LoRA during inference. Metrics were collected on an NVIDIA L40 GPU using BF16 precision.

### 4.4.1 BONUS: EFFECT ON NATURAL RESPONSE LENGTH

`Caprese` implicitly encourages brevity, often even producing shorter responses than from the full thinking model. Intriguingly, this behavior arises despite any enforcement or regularization on response lengths anywhere during training or inference (except for cutting off generation at 32K tokens). Shown in Figure 4 with MATH-500, the shortest mean response length for all problem difficulties is either the full model or `Caprese`. Meanwhile, CATS consistently outputs the longest responses, averaging roughly 1K more across every sample. It is also interesting to note that with increasing problem difficulty, all models and methods naturally allocate more inference tokens towards answering the question. Given CATS and `Caprese` (CATS) achieve similar MATH-500 accuracies to the full DS-Qwen 7B and 14B models, the ability of `Caprese` to cut down the lengthy responses of CATS down to the response lengths of the original model or shorter without compromising performance is a direct memory and latency benefit. This brevity of `Caprese` is more pronounced in larger models, proportionally, from a 5.8% reduction in tokens compared to the full DS-Qwen 7B model to an 8.5% reduction compared to the full DS-Qwen 14B model. The cause of this phenomenon is of interest for future work. See Appendix E for similar observations with GRIFFIN.

## 5 CONCLUSION

To combat the inefficiency of the long and brittle generation process associated with math reasoning, we introduce `Caprese`, a highly performant and efficient method with strong reasoning and language capabilities that is compatible with a broad class of efficient FF algorithms. In the future, it would be interesting to evaluate its benefit in other reasoning domains, explore the use of input-dependent low-rank layers, and investigate the knowledge contained in these low-rank layers. `Caprese`, along with future developments, pushes towards the long-term goal of degradation-free efficient LLM inference.

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

## A    EFFECT OF RANK & SPARSITY

Here, we ablate the relationship between varying ranks in `Caprese` and sparsity levels in CATS. Using Llama 3.2 1B Instruct, we show the test performance of MATH in Figure 5. The same training procedure and data are used as outlined in Section 3. For all ablated ranks, `Caprese` consistently outperforms pure CATS by a large margin when CATS performs poorly relative to the full model. With a couple of exceptions (perhaps due to randomness in the generation process), greater performance is correlated with higher rank.

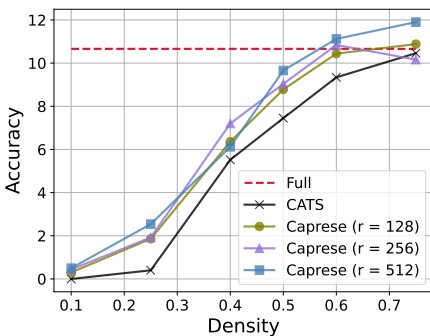

Figure 5: Llama 3.2 1B Instruct's performance on MATH with varying densities of CATS and ranks in `Caprese` with end-to-end training.

## B    CAPRESE PARAMETERS

`Caprese` has a tiny parameter footprint. In Figure 6, we see that `Caprese` adds roughly 1% new parameters relative to the full model with a trend downwards as model size increases.

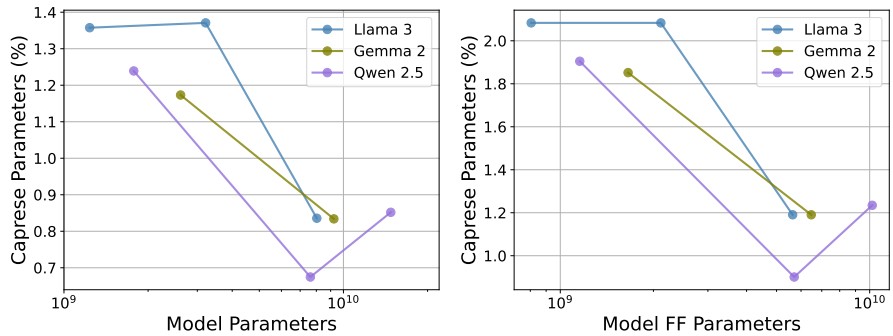

Figure 6: The percent of new parameters that `Caprese` ($r = 256$) adds relative to the entire model (left) and relative to only the FF parameters (right) for the Llama 3, Gemma 2, and Qwen 2.5 model families.

## C    LoRA PERFORMANCE

We report the performance of using LoRA in place of `Caprese` while keeping parameter count constant in Table 5. `Caprese` achieves the highest accuracy in slightly more cases than LoRA, but as per our discussion on LoRA in Section 3.5 and efficiency analysis in Section 4.4, LoRA's main benefit is efficient training, not efficient inference.

Table 5: 0-shot accuracies on mathematical reasoning (GSM8K and MATH) and language generation tasks (CoQA, QASPER, XSum, and CNN/DailyMail) with LoRA. FF sparsity is set to 50%, $r = 256$, and LoRA ranks are set to match the number of parameters that `Caprese` adds.

| Model | GSM8K | MATH | CoQA | QASPER | XSum | CNN/DailyMail |
|---|---|---|---|---|---|---|
| *Llama 3.2 1B Instruct* | 22.44 | 10.66 | 55.43 | 14.43 | 21.65 | 25.60 |
| GRIFFIN | 7.13 | 5.42 | 56.05 | 14.11 | 21.13 | 25.47 |
| Layer-wise LoRA | 8.11 | 5.60 | 56.00 | 14.65 | 21.08 | 25.51 |
| E2E LoRA | 20.47 | **10.10** | 56.10 | 14.41 | 21.09 | 25.62 |
| Layer-wise `Caprese` | 13.72 | 6.62 | 56.07 | 13.40 | 20.65 | 26.18 |
| E2E `Caprese` | **21.00** | 8.44 | 56.55 | 13.88 | 20.71 | 26.18 |
| *Llama 3.2 3B Instruct* | 51.55 | 14.32 | 63.95 | 12.45 | 23.22 | 26.20 |
| GRIFFIN | 28.96 | 10.98 | 64.52 | 12.52 | 22.09 | 25.49 |
| Layer-wise LoRA | 26.46 | 11.28 | 63.88 | 12.55 | 22.12 | 25.96 |
| E2E LoRA | 42.91 | 16.50 | 64.67 | 12.63 | 21.81 | 26.05 |
| Layer-wise `Caprese` | 40.18 | 13.70 | 64.33 | 11.60 | 21.56 | 25.90 |
| E2E `Caprese` | **44.66** | **16.96** | 64.83 | 12.35 | 21.26 | 26.04 |
| *Gemma 2 2B Instruct* | 51.02 | 16.06 | 63.77 | 10.96 | 22.17 | 26.01 |
| GRIFFIN | 33.74 | 11.32 | 63.28 | 11.07 | 18.27 | 22.24 |
| Layer-wise LoRA | 39.58 | 14.66 | 62.98 | 10.54 | 21.58 | 26.64 |
| E2E LoRA | 44.66 | **16.71** | 63.03 | 10.81 | 22.45 | 26.35 |
| Layer-wise `Caprese` | 42.53 | 12.32 | 63.77 | 10.75 | 21.63 | 26.37 |
| E2E `Caprese` | **48.14** | 13.70 | 63.37 | 11.05 | 22.48 | 27.16 |

# D ADDITIONAL INSTRUCT MODELS

In Table 6, we show the performance of `Caprese` in larger instruct models. We again see that `Caprese` obtains the highest accuracy, often recovering the full performance. E2E and layer-wise `Caprese` appear to work better for Llama 3.1 8B Instruct and Gemma 2 9B Instruct, respectively. CATS and `Caprese` considerably improve GSM8K performance of Llama 3.1 8B Instruct, which has trouble following the prompt template with 0-shot.

Table 6: Llama 3.1 8B Instruct's and Gemma 2 9B Instruct's 0-shot accuracies on mathematical reasoning (GSM8K and MATH) and language generation tasks (CoQA, QASPER, XSum, and CNN/DailyMail). FF sparsity is set to 50%, and $r = 256$.

| Model | GSM8K | MATH | CoQA | QASPER | XSum | CNN/DailyMail |
|---|---|---|---|---|---|---|
| *Llama 3.1 8B Instruct* | 27.90 | 13.94 | 63.88 | 15.16 | 21.97 | 25.98 |
| GRIFFIN | 17.44 | 6.16 | 63.37 | 12.63 | 21.53 | 25.89 |
| Layer-wise `Caprese` | 27.14 | 9.72 | 65.05 | 14.21 | 21.92 | 26.31 |
| E2E `Caprese` | **40.49** | **12.64** | 65.50 | 15.35 | 22.05 | 26.42 |
| CATS | 40.56 | 11.80 | 58.85 | 12.26 | 21.43 | 25.67 |
| Layer-wise `Caprese` | 37.68 | 13.66 | 63.92 | 14.34 | 22.58 | 25.79 |
| E2E `Caprese` | **51.86** | **13.88** | 64.27 | 14.42 | 22.08 | 26.13 |
| *Gemma 2 9B Instruct* | 78.17 | 27.64 | 63.78 | 9.91 | 23.98 | 26.46 |
| GRIFFIN | 59.21 | 25.22 | 63.82 | 10.14 | 24.66 | 26.77 |
| Layer-wise `Caprese` | **76.72** | **25.84** | 64.20 | 9.82 | 24.88 | 26.88 |
| E2E `Caprese` | 76.65 | 25.30 | 64.42 | 9.92 | 24.89 | 25.47 |
| CATS | 76.50 | 27.32 | 64.37 | 9.78 | 24.74 | 26.64 |
| Layer-wise `Caprese` | **77.18** | **28.16** | 64.52 | 10.46 | 24.54 | 26.45 |
| E2E `Caprese` | **77.18** | 28.00 | 64.87 | 10.02 | 24.65 | 26.33 |

# E    GRIFFIN RESPONSE LENGTHS

Complementing Figure 4 for CATS, we show the natural response lengths to MATH-500 samples in Figure 7. Here, we see the similar observations as in Section 4.4.1, though the difference between `Caprese` (GRIFFIN) and the full model is slightly larger but decreasing with model size. Even so, response lengths of `Caprese` is still significantly shorter than GRIFFIN.

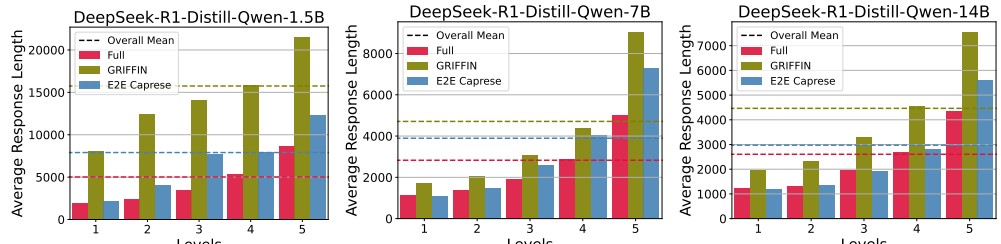

Figure 7: Average number of response tokens for MATH-500 queries with increasing problem difficulty. The global averages are indicated by the dashed lines. Sparsity is set at 50%.

# F    RANK OF LEARNED PARAMETERS

In Figure 8, we plot the relative singular values for each learned product $LR$ from (7) in Llama 3.2 3B Instruct. Although we set the inner rank to be 256, we see that some layers in `Caprese`(CATS) are still very low rank. In comparison, `Caprese`(GRIFFIN) layers are relatively high rank, likely to due to the fact that GRIFFIN performs highly structured FF sparsification. From this, we hypothesize that GRIFFIN may better utilize an increased `Caprese` inner dimension.

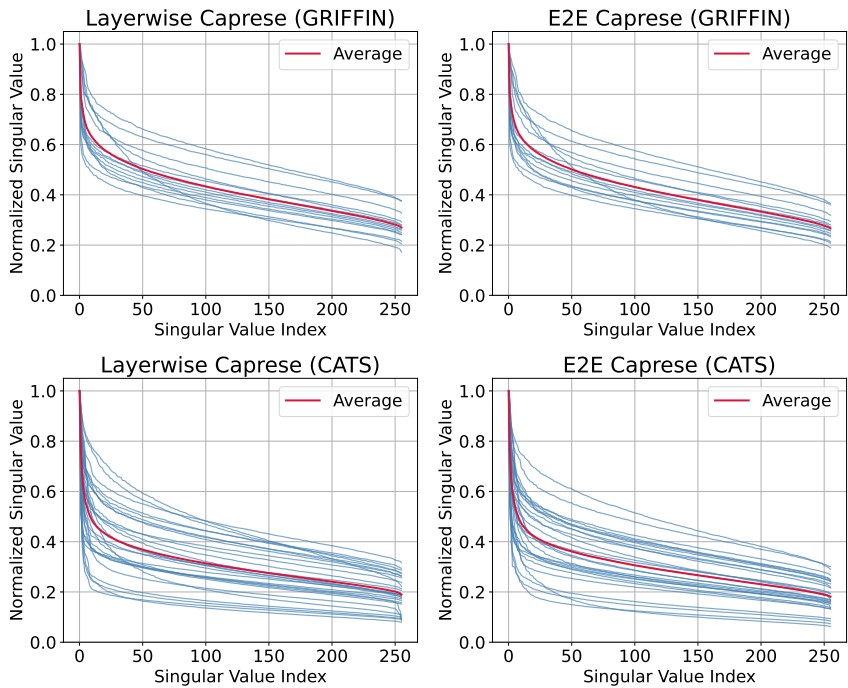

Figure 8: Relative singular values of learned Llama 3.2 3B Instruct `Caprese` layers. Blue lines are individual layers; red lines are averaged across layers.

# G    TRAINING HYPERPARAMETERS

Table 7 lists the hyperparameter settings for training `Caprese` layers. The E2E learning rates lie in the interval [4e-6, 2e-4], where larger models tend to learn better with smaller learning rates.

Table 7: `Caprese` layer-wise and E2E training hyperparameters.

|  | Layer-wise | End-to-end |
| --- | --- | --- |
| Optimizer | Adam | Adam |
| Learning rate | 1e-3 | [4e-6, 2e-4] (varies) |
| Batch size | 128 | 16 |
| Epochs | 20 | 3 |
| Training samples | 2e5 | 2e5 |
| Scheduler | Linear | Linear |
| Warmup | 2% | 2% |

