# OpenReview forum: "Scalable LLM Math Reasoning Acceleration with Low-rank Distillation"
_ICLR.cc/2026/Conference — ICLR 2026 Conference Withdrawn Submission_

### Official Review · Reviewer_3B13 · 2025-10-22

**Soundness:** 4
**Presentation:** 3
**Contribution:** 2
**Rating:** 4
**Confidence:** 3

**Summary:**

This paper introduces Caprese, a method to recover model performance after pruning / efficiency adaptions, by training a LoRA-like adapter to minimize the residual error. Specifically the authors use CATS and GRIFFIN as FF sparsification methods, and show that Caprese can help to regain some of the lost Math performance with minimal computational overhead as the additional linear layers can be fused with existing ones that run in parallel. The distillation is data and memory efficient, just learning small adapters. The authors propose a reselection procedure on top that can improve the performance of Caprese in combination with GRIFFIN.

**Strengths:**

- Recovering strong math performance
- Minimal inference overhead
- Fast training
- Reselection procedure for improved performance similar to speculative decoding
- Analysis of the response length of efficiency-optimized models vs. original

**Weaknesses:**

- Similarity to LoRA, LESS just applied to FF layers resulting in a lack of novelty
- Not reproducible given private training set

**Questions:**

- What is "a common synthetic math training set"?
- How do you expect this to work around attention layers, e.g. when using sparsified / pruned attention, in the reasoning case?
- Your analysis on average response length is quite important. In the reasoning setting with a late final answer, wouldn't it make sense to compare overall runtime as efficiency metric - this way GRIFFIN / Caprese-GRIFFIN would underperform, despite per-token speedup, right? Could you add an analysis on this overall cost / runtime?

---

> ### Author Response · Authors · 2025-11-21
>
> Thank you for your comments on our paper! We answer each point you brought up below. Let us know if we have addressed them appropriately.
>
> **Concerns**
>
>  - While LoRA, LESS, and Caprese all use low-rank structure in different parts of the model, they are very distinct. LoRA focuses on efficient training, not inference. The formulation of LESS prevents added weights from being merged with existing weights, meaning computation is sequential. Moreover, LESS is primarily a reduction of the token dimension. In contrast, Caprese is efficient during training and inference while reducing the inner dimension with mergeable weights. A brief summary is included in Table R2.
>
> - We use data from [1].
>
> Table R2: LoRA vs. LESS vs. Caprese
>
> || LoRA|LESS|Caprese|
> |-|-|-|-|
> |Reduced Training Cost|yes|yes|yes|
> |Reduced Inference Cost|no|yes|yes|
> |Mergeable with Existing Weights|yes|no|yes|
> |Hidden Dimension Reduction|no|no|yes|
>
> **Questions**
>
> - See above bullet point.
>
> - First, we’d like to highlight that Caprese is not restricted to reasoning. We mainly test on reasoning since we see the most degradation by previous sparse methods. Second, for sparse and pruned attention blocks, it may be possible to do something similar. Recall that a major appeal for Caprese is the ability to perform computation with added parameters in parallel with existing parameters, so we would need to find a way to do this in attention layers. Since Caprese is a compression along the feature dimension axis, this could be composable with methods that reduce along the token axis like sparse KV caching.
>
> - This is an interesting idea. We show results of this combined metric in Table R3, separated into problem difficulty in MATH-500. We see a general improvement in latency for Caprese, whereas GRIFFIN, although fast per-token, needs to generate many more tokens before completion.
>
> Table R3: Percent change in latency compared to the full DS-Qwen 14B model when combining natural generation completion in MATH-500 and per-token latency. Negative values (better) indicate reductions (improvement) in latency.
>
> ||Level 1|Level 2|Level 3|Level 4|Level 5|Avg|
> | - |:-:|:-:|:-:|:-:|:-:|:-:|
> GRIFFIN|32.79%|48.11%|39.40%|39.99%|43.99%|40.86%
> Caprese|-18.84%|-12.35%|-18.85%|-13.23%|6.72%|-11.31%
>
>
> In addition, perhaps of interest to you, we have added 3 new generative tasks: XSum (summarization), CNN/DailyMail (summarization), and GPQA (graduate-level science questions). You can find these new results in Tables 1, 2, 5, and 6 of our revised paper.
>
>
> [1] Mitra, A., et al. Unlocking the potential of slms in grade school math, 2024.

---

### Official Review · Reviewer_Dj7H · 2025-10-31

**Soundness:** 3
**Presentation:** 3
**Contribution:** 2
**Rating:** 4
**Confidence:** 4

**Summary:**

This paper introduces an efficient method of distilling models, with a primary focus on adapting the feed-forward blocks. Despite its relatively low budget, the method achieves promising results in almost fully recovering the mathematical capabilities of the distilled models compared to the teacher model. More exactly, Caprese recovers substantial performance with a small training budget (only 20K synthetic samples) and minimal overhead (roughly 0.8% additional parameters).

**Strengths:**

- One strength for the Caprese method is the efficiency compared to baselines, achieving near teacher-level performance on challenging benchmarks like MATH and GSM8K with a minimal training budget. This trade-off between computational cost and capability is a significant practical contribution, making high-performing, specialized models more accessible for real-world deployment.
- The strategy of performing low-rank distillation specifically on the FFN blocks to learn the residual necessary for complex tasks is an original and effective architectural choice.
- Accordingly, the paper clearly explains the proposed mechanism and its implementation. The methodology is straightforward and the experiments generally are well-structured.

**Weaknesses:**

- The primary focus and strongest empirical evidence are on mathematical reasoning tasks. It remains unclear how the method would transfer effectively to a diverse set of other domains, such as code generation or more complex general language understanding tasks.
- While the authors claim "strong" language capabilities, the empirical evidence is restricted to only a few benchmarks (CoQA, QASPER). On benchmarks like QASPER, the Caprese method does not consistently outperform the established GRIFFIN baseline (e.g., Table 1). The evaluation of language capabilities should be extended to support the strong claim.
- The initial performance of the original teacher models (e.g., Llama 3.1 8B Instruct GSM8K: 27.90, MATH: 13.94) appears unusually low for a model of that scale. This raises a question about the specific prompting strategy or evaluation setup used for the teacher model versus the student model during inference. Clarifying this discrepancy is essential for fully evaluating the true performance "recovery."
- The low-rank size of 256 is considered as being low-cost, but a small ablation study showing the sensitivity of the performance recovery to the rank dimension would significantly strengthen the analysis of the method's trade-off between parameter overhead and performance gains.

**Questions:**

1. Could you please elaborate on the surprisingly low performance of the Llama 3.1 8B Instruct teacher model on the GSM8K and MATH benchmarks as shown in Table 6? Specifically, is the observed performance gain of Caprese potentially related to the specific prompts or few-shot settings used during the benchmark evaluation for the student vs. teacher models?

2. Please provide a small analysis on the effect of the batch size for the distillation setting. How does the efficiency of your method scale, and how effective is it to use Caprese for a larger batch size during training compared to conventional methods?

3. The claim of "strong [...] language capabilities" requires more evidence. Could you please justify the focus on only CoQA and QASPER for language tasks? I strongly suggest extending the evaluation to a more diverse set of standard LLM benchmarks, such as MMLU, DROP, or a selection from the HELM suite, to substantiate this claim.

4. Would it be possible to include a comparison for an additional domain? Even a small experiment, perhaps on a code generation task (e.g., HumanEval), would be beneficial. This would demonstrate that, even if the primary focus is math, the method does not perform drastically worse than the teacher on a different, yet still STEM-related, task.

---

> ### Author Response · Authors · 2025-11-21
>
> Thank you for taking the time to review our paper! Since some of your comments and questions have overlap, we group them by topic below. Please let us know if we have addressed them appropriately.
>
> **Other Tasks**
>
> Thank you for the suggestions, we have added 3 new generative tasks: XSum (summarization), CNN/DailyMail (summarization), and GPQA (graduate-level science questions). You can find these new results in Tables 1, 2, 5, and 6 of our revised paper. These new results further reinforce our method’s benefits. We see no degradation in these new language tasks in every model. We also see nearly full recovery of GPQA performance by Caprese, whereas CATS/GRIFFIN severely degrade accuracy, which shows our method’s generalizability beyond math despite being only trained on math. We would like to emphasize that the high accuracy of GRIFFIN/CATS on language tasks is expected. The main purpose of including language tasks is to show that Caprese does not degrade language capabilities even though we significantly recover math performance. Therefore, the takeaway from these language tasks is that all these methods perform equivalently to the full model, and their differences are mainly highlighted in non-language tasks.
>
>
> **Llama 3.1 8B Instruct Performance**
>
> All instruct models are tested with identical 0-shot prompts per task, which can explain the relatively low performance of the original Llama 3.1 8B Instruct performance. However, upon inspecting the generated outputs for GSM8K, we noticed that Llama 3.1 8B Instruct had difficulty following instructions to denote their final answer (it often did not place its answer after “####”). Hence, we reran GSM8K on Llama 3.1 8B Instruct with the same prompt and text extraction as MATH (answers in “\boxed{}”) and show the improved results in Table R1. We again see the benefit of Caprese with this alternative prompt. We can add this discussion on prompt sensitivity to the next version of our paper.
>
> Table R1: GSM8K rerun with new prompt.
>
> | Model|GSM8K|
> | - |:-:|
> |Llama 3.1 8B Instruct| 53.98
> |
> |GRIFFIN|20.47
> |Layer-wise Caprese|37.60
> |E2E Caprese|51.40
> |
> |CATS|50.95
> |Layer-wise Caprese|51.93
> |E2E Caprese|58.00
>
>
> **Caprese Rank**
>
> The rank of 256 is low-cost compared to the inner dimension of FF blocks which is typically at scale of (10^5). Hence, adding Caprese rank to this is rather insignificant. You can find a brief exploration on the relationship between rank and performance in Appendix A.

---

> > ### Comment · Reviewer_Dj7H · 2025-11-26
> >
> > Thank you for your answer and for addressing my concerns. I am willing to raise my score to a 6.

---

### Official Review · Reviewer_bFEe · 2025-11-01

**Soundness:** 2
**Presentation:** 2
**Contribution:** 2
**Rating:** 2
**Confidence:** 3

**Summary:**

Caprese recovers math reasoning performance lost from sparse FF methods (GRIFFIN, CATS) by distilling residuals into rank-256 low-rank layers (~1% parameters) trained on 20K synthetic samples. Results show full recovery (DS-Qwen 7B: 62.5%→78.1% AMC) without harming language tasks, preserving efficiency (~2B parameter cut, >16% latency reduction). Works across instruct/thinking models, scales with Best-of-N, and unexpectedly reduces output length by 5-8.5%.

**Strengths:**

**Strong Empirical Evidence**: Sparse FF methods devastate math (Gemma 2 2B: 51%→34% GSM8K) despite working well on language. Figure 2's oracle analysis shows rank-256 approximation reduces residual error more than doubling sparsity density, motivating low-rank design.

**Comprehensive Experiments**: Robust across 1B-32B models (Llama, Gemma, Qwen, DeepSeek), math/language tasks, and test-time scaling. Figure 3 shows +7% Pass@100 vs. full model with 15.8% less compute. Table 2 demonstrates recovery on 32K-token thinking models.

Only 20K samples, 23 epochs, frozen weights. Figure 6 shows ~1% parameter overhead. Reselection (Table 3) improves DS-Qwen 1.5B from 34%→59% AMC by updating GRIFFIN metrics. Unexplained brevity (Figure 4): 5-8.5% fewer tokens than full models despite no length penalties.

**Weaknesses:**

**No Mechanistic Understanding**: What do L,R matrices capture? No SVD analysis, probing, or visualization. Why does layer-wise work better for Gemma but E2E for Llama (Table 6)? No ablation isolating layer-wise vs. E2E contributions.

**Narrow Scope**: Claims "any task" but only math + 2 language QA tasks. Missing: code generation, creative writing, dialogue, multi-modal. No baselines: full→sparse KD, error-focused training, adaptive rank

**Questions:**

1. Does Caprese work with MoEs, pruning, quantization? Or only activation-based sparsity?
2. Provide SVD analysis of L,R, visualization of captured components, and ablation isolating layer-wise vs. E2E.
3. Why shorter responses? Is it filtering redundancy, implicit regularization, or correlation with error reduction?
4. Compare to full→sparse KD, LoRA (main text), and more training data for sparse methods.

---

> ### Author Response · Authors · 2025-11-21
>
> Thank you for the pointers to improve our work! We have taken actions based on your suggestions, which we address in order below. Please let us know if you have additional questions.
>
> **Mechanistic Understanding**
>
> Thank you for your interest in the inner workings of our method. We have included an analysis on learned weights in Appendix F of our revised paper. In it, we show the rank of LR and observe that Caprese(GRIFFIN) parameters are much higher in rank than learned Caprese(CATS) parameters. This is likely due to the fact that GRIFFIN is a much more structured algorithm, so it is easier for Caprese to capture the residuals.
>
> As for the intuition behind E2E vs. Layer-wise, we empirically observe that E2E usually attains greater accuracy, but with larger models (e.g., DS-Qwen-14B), layer-wise trained layers tend to preserve performance better. This could be a consequence of overfitting when fine tuning E2E. This is in fact good news for large models since layer-wise training is much more efficient, as all layers can be trained in parallel.
>
> **Scope**
>
> We have expanded our tasks to include summarization (XSum and CNN/DailyMail) and science (GPQA) which have been added to the revised paper’s Tables 1, 2, 5, and 6. These new results reinforce our contribution. Like previous tasks, we see all methods preserve summarization quality, but sparse-only methods severely harm GPQA which Caprese successfully prevents.
>
> **Questions**
>
> - Caprese pairs well with any method that lets us integrate new weights into existing weights for parallel execution, meaning it would work for any of the techniques you have listed. For MoEs, Caprese parameters would be analogous to a linear expert that is always activated. For pruning and quantization, you can append these new weights exactly like how we did in this paper.
>
> - See the above bullet point on Mechanistic Understanding.
>
> - We qualitatively observe fewer instances of mistakes and backtracking when compared to sparse-only models. To separate the correlation with accuracy, we can see that in the right-most figure of Figure 4, we still see vast length differences despite each method’s performance on MATH500 being within 2% of each other.
>
> - We include a comparison with LoRA in terms of performance (Table 5) and efficiency (Table 4). While the listed methods including LoRA are great for efficient training, our main focus is on efficient inference, so we compare Caprese to inference methods. Efficient training is a byproduct of Caprese’s design, but its core focus is on accelerating generation which these other methods do not.

---

> > ### Comment · Reviewer_bFEe · 2025-11-23
> >
> > Thank you for your detailed responses. I appreciate the clarifications, particularly regarding the SVD analysis of L,R matrices and the distinction between layer-wise and end-to-end training. These additions helped deepen my understanding.
> >
> > I believe further testing on more challenging tasks, such as AIM E 25, Olympiad, Minerva, or MATH500, would provide stronger evidence for Caprese's broader applicability, especially in advanced reasoning tasks.
> >
> > Additionally, other reviewers raised concerns about overclaiming in the original version. I suggest refining the language to better align the claims with the presented evidence.
> >
> > Thanks again for your efforts, and I look forward to the next version. I have raised my score to 4.

---

> > > ### Author Response · Authors · 2025-11-23
> > >
> > > Thank you for your response!
> > >
> > > We have 5 challenging tasks (MATH500, AIME, AMC, BRUMO, and GPQA) using reasoning models in Table 2 which demonstrates strong recovery in long generation with reasoning models.
> > >
> > > As for wording, the main point brought up by Reviewer Dj7H was to include more evidence of Caprese's "strong language capabilities". Based on their suggestion, we added new language tasks (XSum and CNN/DailyMail) to Tables 1, 5, and 6 in the revision which further back up this claim. We would like to emphasize the main purpose of including language tasks is to show that Caprese does not degrade language capabilities even though we significantly recover math performance. Therefore, the takeaway from these language tasks is that all these methods perform equivalently to the full model, and their differences are mainly highlighted in non-language tasks. Of course, we will go through the paper again to refine and support any of the claims we make.
> > >
> > > Thank you again and please let us know what you think!

---

### Note · Authors · 2025-12-03

I have read and agree with the venue's withdrawal policy on behalf of myself and my co-authors.